# Age Is Just a Number: Progress and Obstacles in the Discovery of New Candidate Drugs for Sarcopenia

**DOI:** 10.3390/cells12222608

**Published:** 2023-11-11

**Authors:** Hyun-Jun Kim, Da-Woon Jung, Darren Reece Williams

**Affiliations:** New Drug Targets Laboratory, School of Life Sciences, Gwangju Institute of Science and Technology, Gwangju 61005, Republic of Korea; khjun9013@gist.ac.kr

**Keywords:** sarcopenia, drug discovery, skeletal muscle, aging

## Abstract

Sarcopenia is a disease characterized by the progressive loss of skeletal muscle mass and function that occurs with aging. The progression of sarcopenia is correlated with the onset of physical disability, the inability to live independently, and increased mortality. Due to global increases in lifespan and demographic aging in developed countries, sarcopenia has become a major socioeconomic burden. Clinical therapies for sarcopenia are based on physical therapy and nutritional support, although these may suffer from low adherence and variable outcomes. There are currently no clinically approved drugs for sarcopenia. Consequently, there is a large amount of pre-clinical research focusing on discovering new candidate drugs and novel targets. In this review, recent progress in this research will be discussed, along with the challenges that may preclude successful translational research in the clinic. The types of drugs examined include mitochondria-targeting compounds, anti-diabetes agents, small molecules that target non-coding RNAs, protein therapeutics, natural products, and repositioning candidates. In light of the large number of drugs and targets being reported, it can be envisioned that clinically approved pharmaceuticals to prevent the progression or even mitigate sarcopenia may be within reach.

## 1. Introduction: Skeletal Muscle Aging, Definition of Sarcopenia and Current Treatment Options

Skeletal muscle is the major organ system in the body and constitutes around 40% of total body weight [1]. The main functions of skeletal muscle related to contraction include the production of movement, maintenance of posture, stabilization of joints, and respiration [2] letal muscle also plays a central role in metabolic health, because it is a major site of carbohydrate and fatty acid metabolism [3] keletal muscle is the main protein reservoir in the body and provides amino acid substrates for glucose and glycogen synthesis [4]. More recently, the endocrine functions of skeletal muscle have been recognized. Through the action of myokines (cytokines synthesized and released during contractions) and myometabolites (such as lactate, ammonia, and adenosine), skeletal muscle acts as a sensor of nutrient status, environmental challenges, and organelle dysfunction that can regulate systemic nutrient and stress signaling [5]. Consequently, skeletal muscle can influence the physiology of multiple organs, such as the brain, liver, and gut.

A remarkable feature of skeletal muscle is its plasticity. Myofibers and associated tissues, including capillaries and motor neurons, adapt to various stimuli, such as contractions, loading, nutrition, and environmental factors [6]. These adaptations in myofibers involve alterations to contractile properties, mitochondrial function, metabolic capacities, and both intercellular and intracellular signaling pathways [6]. Skeletal muscle is also sensitive to aging (Figure 1). The most prominent changes are reduced mass and strength [7]. Aging-related skeletal muscle atrophy typically begins in the fourth decade of life, with a loss of 3–8% muscle mass per decade thereafter [8]. This decline begins to accelerate after the age of 60 [9,10]. By 80 years of age, there is an approximately 30–50% reduction in skeletal muscle mass [11]. One important consideration of skeletal muscle aging is that strength declines more rapidly than mass [12]. For example, the reduction in knee extension strength in old age is two- to four-times greater than the loss of muscle mass [13]. Overall leg strength decreases 10–15% per decade until 70 years of age, at which time it is approximately 20–40% lower compared to young adults. After 70 years, this decline in strength also accelerates to a 25–40% strength reduction for each decade [6].

Although reduced mass and function are the two hallmarks of skeletal muscle aging, there are multiple structural and biochemical changes in the muscle tissue that contribute to this decline [14]. The myofiber cross-sectional area (CSA) is decreased, and this is accompanied by an increased accumulation of fat tissue [15]. The myofiber composition also changes, with declining numbers of type II ‘fast’ twitch myofibers associated with strength that rely on anaerobic glycolysis for energy production, alongside increasing numbers of type I ‘slow’ twitch myofibers associated with endurance that rely on oxidative phosphorylation for energy production [16]. There is also a decline in the numbers and activity of motor neurons, especially alpha-motor neurons that innervate type II myofibers [17]. Additionally, the denervated type II myofibers may be re-innervated by the branching of adjacent motor neurons that innervate type I myofibers, resulting in their eventual conversion into type 1 myofibers [18]. Mitochondria also become dysfunctional in aged skeletal muscle [19]. Human studies have reported reduced mitochondria size, biogenesis, respiration rate, and increased accumulation of mutations in the mitochondrial genome [20]. Other changes in aged skeletal muscle include a higher number of myonuclei undergoing apoptosis, elevated levels of oxidative stress, increased production of proinflammatory cytokines (known as ‘inflammaging’), and a reduction in the regenerative potential and self-renewal capacity of skeletal muscle stem cells (also described as satellite cells due to their proximal location to mature myofibers) [21].

Aging-related skeletal muscle atrophy was given the term sarcopenia by Irwin Rosenberg in 1989 (derived from the Greek σάρξ sarx, “flesh” and πενία penia, “poverty”) [22]. The definition and diagnosis of sarcopenia have evolved over time. In 2010, the European Working Group on Sarcopenia in Older People (EWGSOP) provided a clinical diagnosis of sarcopenia based on the loss of skeletal muscle mass and function [23]. In 2016, sarcopenia was designated as a disease by the World Health Organization (ICD-10 code M62. 84: soft tissue disorders) [24]. In 2018, the EWGSOP2 updated their clinical diagnosis of sarcopenia to increase focus on reductions in muscle function (strength) and severe sarcopenia involving the loss of physical function [25]. In 2020, the Asian Working Group of Sarcopenia (AWGS) defined its clinical criteria for sarcopenia based on poor physical function in the presence or absence of reduced muscle mass [26]. The Foundations of National Institutes of Health (FNIH) sarcopenia project in the USA incorporated clinically relevant cut-off points of both low muscle mass and strength [27]. However, there are no universal, standardized parameters for diagnosing sarcopenia, with health organizations in different continents designating their own criteria [27]. For example, the EWGSOP definition includes grip strength of <30 kg for men and <20 kg for women, whereas the AWGS definition includes grip strength <26 kg for men and <18 kg for women [27,28,29].

These differences in the criteria used to define sarcopenia influence calculations of its prevalence in the aged population [27]. The ethnicity of the population studied and healthcare setting (community versus hospital) also affect estimates of sarcopenia prevalence [27,30]. A systematic review by Cruz-Jentoft et al. in 2014 reported a prevalence rate of up to 29% in community-dwelling older people aged ≥60 years and up to 33% in long-term care aged >70 years [31]. A recent systematic review plus meta-analysis by Petermann-Rocha et al. reported a prevalence of 8–36% for <60 years and 10–27% for ≥60 years [32]. Overall, the prevalence of sarcopenia in the general population is predicted to increase due to gains in worldwide life expectancy and decreased fertility [33]. For example, in the USA in 2030, the population over the ages of 65 and 85 is predicted to rise by 147% and 389% compared to 2018 [33].

Sarcopenia is associated with an increased risk of developing a range of negative health sequelae, such as poor overall and disease progression-free survival rate, postoperative complications, longer hospitalization in patients with different medical issues, a greater propensity for falls and broken bones, metabolic diseases, cognitive impairment, (such as Alzheimer’s disease), and increased mortality [34]. Although aging is the major risk factor for developing sarcopenia, it should be noted that other factors also increase the risk, such as physical inactivity, malnutrition, smoking, extreme sleep duration, and diabetes [34].

The current treatment options for sarcopenia are based on exercise programs and nutritional supplements. At the time of writing, there is no clinically approved drug for this disorder. Resistance exercise (RE) has been shown to improve skeletal muscle mass and strength in patients with sarcopenia [35]. The majority of exercise regimes prescribed involve a combination of balance, aerobic, and RE, which have been shown to be more effective than RE alone [36]. Full body exercise programs should be performed at least twice per week, with a relatively high level of effort [37]. Nutritional supplements, such as greater protein intake, are used to improve the muscle hypertrophic response to RE [38]. Balanced meal frequency plus protein intake has been shown to increase muscle protein synthesis in the elderly [39]. Animal-derived proteins, such as whey protein, are recommended due to containing a wider spectrum of amino acids and increased ease of digestion compared to plant-derived proteins [40]. The branched-chain amino acids (leucine, isoleucine, and valine; BCAA) are important for sarcopenia research, because they have been shown to promote protein synthesis/turnover and glucose metabolism in skeletal muscle [41]. A recent meta-analysis indicated that BCAA supplementation may have beneficial effects on muscle mass and strength in the elderly [42]. Other nutrition-based supplements for treating sarcopenia include beta-hydroxy-beta-methylbutyrate, a naturally occurring compound in humans that is derived from leucine and was shown to preserve or increase strength in subjects with aging-related muscle atrophy [43]. Vitamin D supplementation has also shown potential to treat and prevent sarcopenia [44]. More generalized dietary interventions, such as the Mediterranean diet, have been reported to delay sarcopenia progression in elderly communities [45].

Although exercise and nutritional interventions have shown effectiveness for treating sarcopenia, there is still a need to develop therapeutics. For example, aging-related skeletal muscle loss is still observed in the elderly that follow healthy lifestyles, such as Master’s Athletes [46]. Adherence to exercise and nutrition interventions can also be problematic [47], especially in patients with dementia [47]. Beneficial responses to exercise and nutrition may also be blunted by anabolic resistance and inefficient digestion [48,49].

Numerous drug candidates have been assessed in clinical trials. Recent examples include perindopril (angiotensin-converting enzyme (ACE) inhibitor) combined with leucine [50], anamorelin hydrochloride (gherlin receptor agonist) [51], water extract of *Codonopsis lanceolata* (flowering plant native to East Asia) [52], and bimagrumab (activin receptor type-IIB (ActRIIB) inhibitor) [53]. Although there is currently no clinically approved drug for sarcopenia, much progress has been made in characterizing the cellular mechanisms underpinning the development of sarcopenia (Figure 2 and reviewed in [54]), and this has led to the discovery of many new drug candidates. This review will focus on these recent advances in the pre-clinical development of these novel candidate drugs and biological targets for the treatment of sarcopenia.

## 2. Novel Candidate Drugs under Pre-Clinical Development

### 2.1. Mitochondria-Targeting Compounds

As the ‘powerhouses’ fueling contraction, the integrity of the mitochondrial network has a critical influence on skeletal muscle function. Mitochondria in skeletal muscle form a reticulum providing conductive pathways for energy distribution [55]. This reticulum is a dynamic network regulated by fusion proteins, such as mitofusins 1 and 2 (Mfn1 and 2), and fission proteins, such as dynamin-related protein 1 (Drp1) and mitochondrial fission 1 (Fis 1) [56,57,58]. Damaged mitochondria in healthy muscle are degraded in autophagosomes via the process of mitophagy. The mitochondria pool is maintained via biogenesis regulated by master genes, such as peroxisome proliferator-activated receptor gamma coactivator 1-alpha (PGC-1α) [59,60]. These quality control mechanisms fail in aging muscle and dysfunctional mitochondria accumulate, resulting in decreased ATP production, altered proteostasis, reduced calcium handling, elevated oxidative stress, and inflammation, all of which contribute to muscle atrophy [61,62,63]. Therefore, drugs that target dysfunctional mitochondria have the potential to ameliorate sarcopenia progression.

Elamipretide (SS-31) is a tetrapeptide that targets mitochondria to reduce reactive oxygen species (ROS) production and stabilize cardiolipin, which is a major component of the inner mitochondrial membrane [64,65]. Campbell et al. administered SS-31 to aged mice (26 months old; approximately equivalent to 88 years in human age [66]) for 8 weeks via implanted osmotic minipumps [67]. Assessment of the gastrocnemius muscle indicated the restoration of redox homeostasis compared to young mice (5 months old), as shown by the reversal of cysteine S-glutathionylation post-translational modifications across the muscle proteome [67]. Mitochondria quality was improved, and treated mice showed reduced muscle fatigue in a treadmill running test [67]. These findings suggest that the drug targeting of mitochondrial oxidative stress alone may be sufficient to prevent aging-related muscle dysfunction.

BAM15 is an orally available mitochondrial-targeted furazano [3,4-b] pyrazine protonophore that selectively lowers respiratory coupling and protects against diet-induced obesity [68]. Dantas et al. tested BAM15 in a model of sarcopenic obesity (80-week-old mice fed a high-fat diet (HFD) for 10 weeks) [69]. BAM15 increased muscle mass, strength, and locomotor activity [69]. These beneficial effects were due to enhanced mitochondrial function and quality control, along with reduced endoplasmic reticulum (ER) misfolding (indicating lower ER stress) [69].

A number of mitochondria-targeting drugs have shown effectiveness in other models of skeletal muscle atrophy. For example, the cellular levels of NAD nicotinamide adenine dinucleotide (NAD(+)) are important for mitochondria function [70]. Pirinen et al. demonstrated that small-molecule MRL-45696, a pan inhibitor of NAD(+) consuming poly(ADP-ribose) polymerases (PARPs), increased the amount of mitochondrial respiratory complexes, enhanced respiratory capacity, and boosted exercise endurance in adult mice [71]. Improved mitochondrial function was also observed in myotubes derived from obese humans with type 2 diabetes (T2DM) [71].

Mitoquinone mesylate (MitoQ) is a mitochondria-targeting antioxidant [72]. MitoQ is a synthetic analogue of coenzyme Q10 with superior bioavailability and mitochondrial penetration. Pin et al. showed that oral administration of MitoQ to tumor-bearing mice prevented skeletal muscle wasting, improved strength, and downregulated the expression of the wasting-related atrogenes atrogin-1 and MuRF-1 [73]. These beneficial effects occurred without influencing tumorigenesis.

Mito-TEMPOL is a mitochondria-targeting superoxide dismutase mimetic that combines the antioxidant moiety TEMPOL (also known as 4-hydroxy-TEMPO) with the lipophilic cation triphenylphosphonium to facilitate permeability through lipid bilayers and accumulation in mitochondria [74]. Supinski et al. demonstrated that the intraperitoneal (IP) delivery of Mito-TEMPOL was effective in preventing diaphragm atrophy in a murine model of sepsis (critically ill patients, such as those with sepsis, commonly develop intensive care unit-acquired muscle weakness) [75,76]. Mito-TEMPOL treatment improved diaphragm force generation, mitochondrial function, and myosin heavy chain (MHC) content and lowered proteolytic enzyme activities [75]. Furthermore, Liu et al. tested the protective effect of Mito-TEMPOL in a murine chronic kidney disease (CKD) model of uremia-induced skeletal muscle atrophy [77]. Analysis of the gastrocnemius muscle showed that Mito-TEMPOL treatment increased muscle mass, increased mean myofiber CSA, and reduced the expression of atrogenes atrogin-1, MuRF-1, and myostatin ((GDF8), a myokine and member of the transforming growth factor (TGF)-β family that suppresses muscle growth) [77]. Transmission electron microscopy also revealed improvements in mitochondrial dysfunction [77]. These changes were reflected by increased grip strength in the Mito-TEMPOL-treated mice [77]. It should be noted that these beneficial effects of Mito-TEMPOL in this study may also be due to improvements in renal function in addition to direct effects on skeletal muscle.

Although MRL-45696, MitoQ, and Mito-TEMPOL have not been assessed in models of sarcopenia, their effectiveness in other types of skeletal muscle atrophy suggest that they may be attractive candidates for testing in aging models. A number of novel targets related to mitochondria have also been identified that may be suitable for future drug development. For example, Kimoloi et al. demonstrated that a dominant–negative variant of mitochondrial helicase produced muscle atrophy and increased the proportion of cytochrome c oxidase-negative myofibers in aged (24 months old) mice [78]. Histological analysis also revealed the presence of ragged red fibers with clumps of diseased mitochondria accumulated in the subsarcolemmal area [78]. Satellite cell function was severely compromised, and degenerated fibers were replaced with fibrotic and adipogenic tissue [78]. Recently, Koh et al. reported that the expression of PGC-1α4 (an isoform of the master regulator of mitochondrial biogenesis (PGC-1α)) increases in response to RE in humans, regulates myofiber hypertrophy, and enhances glycolysis to supply ATP for muscle contractions [79]. The role of PGC-1α4 in muscle adaptation to RE was observed in both younger and older subjects [79]. Thus, the authors speculated that PGC-1α4 can be a therapeutic target to develop exercise mimetics for sarcopenia patients that cannot participate in RE programs or suffer from anabolic resistance [79].

The studies described above indicate that strategies to reduce oxidative stress and maintain the mitochondrial network in skeletal muscle may be beneficial to prevent aging-related weakness. In this regard, indicators from human studies in research fields such as sports physiology could be useful for selecting compounds that reduce oxidative stress in aging muscle (see Braakhuis and Hopkins for a review [80]). For example, the antioxidant *N*-acetylcysteine has been shown to delay skeletal muscle fatigue in healthy subjects, and intravenous administration in athletes produced improvements in sprint performance [81,82]. In addition, spirulina (an extract of blue-green algae containing numerous antioxidants) is believed to have been consumed by the Cuban and Chinese Olympic teams to enhance their response to training [80,81].

These results from targeting mitochondria (summarized in Table 1) also indicate the importance of glucose metabolism (a mitochondrial function) in the maintenance of skeletal muscle function and the potential negative effects of diabetes. The next section of this review will discuss studies of anti-diabetes drugs in skeletal muscle aging.

### 2.2. Potential Utilizatioin of Anti-Diabetes Drugs for Sarcopenia

Skeletal muscle is a major site of postprandial glucose uptake and an important tissue for the systemic utilization of free fatty acids (FFAs) [83]. Type 2 (adult onset) diabetes (T2DM) is characterized by insulin resistance and high blood glucose and FFA levels [84]. T2DM is associated with skeletal muscle atrophy due to factors such as increased oxidative stress and inflammatory signaling mediated by nuclear factor kappa-light-chain enhancer of activated B cells (NF-κB) and the c-Jun N-terminal kinases [85]. Therefore, it is reasonable to hypothesize that anti-T2DM drugs could be employed as preventative or treatment therapeutics for sarcopenia, because aging is generally associated with increased levels of insulin resistance [86]. Both pre-clinical and clinical data can be used to assess the potential of the different classes of anti-T2DM drugs for sarcopenia treatment.

Metformin (also known as Glucophage, Fortamet) is a first-line drug treatment for T2DM and is generally well tolerated [87]. The mechanism is not fully characterized but is known to involve the AMP-activated protein kinase (AMPK) pathway that also regulates the skeletal muscle response to exercise [88,89]. Clinical data of the effects of metformin on skeletal muscle in T2DM patients have shown mixed results [90]. Musi et al. and Wang et al. showed no effect of metformin on lean body mass in T2DM patients after metformin treatment for 10 weeks or 6 months, respectively [91,92]. In contrast, J R Rodríguez-Moctezuma et al. reported that metformin increased lean mass in patients at risk of developing T2DM [93]. Thus, further experiments are required to clarify the effect of metformin treatment on aging-related muscle wasting and may focus on assessments of skeletal muscle performance (e.g., grip strength or rotarod in mice). As noted above, muscle strength declines more rapidly than mass in sarcopenia. Previous laboratory-based studies suggest that metformin may actually exacerbate aging-related muscle weakness [90]. Hindlet et al. showed that metformin reduced intestinal peptide absorption in mice via downregulation of the H(+)/peptide cotransporter 1 [94]. A recent study by Kang et al. in mice found that metformin reduced muscle fiber CSA and upregulated the expression of myostatin [95]. Interestingly, grip strength was not significantly affected by metformin treatment [95].

Sulfonylurea (such as glibenclamide (Diabeta)) and meglitinide/glinides (such as Prandin and Starlix) are anti-T2DM drugs that work by enhancing insulin secretion from the pancreas via the inhibition of ATP-sensitive K+ (KATP) channels [90,96]. KATP channels are also expressed in skeletal muscle and preclinical evidence suggests that drugs targeting this channel produce muscle atrophy and apoptosis [97,98]. This was also supported by a meta-analysis of the clinical data, which reported a 12-fold higher incidence of muscle atrophy in patients prescribed glibenclamide compared to non-sulfonylureas or glinides [99].

Thiazolidinedione-based drugs, such as pioglitazone (Actos) and lobeglitazone (Duvie), induce glucose uptake from the bloodstream by activating peroxisome proliferator-activated receptor (PPAR) nuclear receptor proteins [90]. Clinical trials reported lower lean leg mass after pioglitazone treatment in patients with prediabetes, along with acute rhabdomyolysis in T2DM patients prescribed pioglitazone or troglitazone [100,101,102]. Thus, thiazolidinediones should be prescribed with caution in patients with aging-related muscle wasting.

Sodium-glucose co-transporter 2 (SGLT2) inhibitors, dipeptidyl peptidase IV (DPP-IV) inhibitors, and glucagon-like peptide-1 (GLP-1) analogs are T2DM medications with conflicting reports on skeletal muscle wasting [90,103]. Sano et al. reported increased grip strength in male and female T2DM patients treated with the SGLT2 inhibitor empagliflozin, whereas a meta-analysis by Pan et al. reported that these drugs reduce skeletal muscle mass [104,105]. In murine studies, the DPP-IV inhibitor vildagliptin was shown to switch lipid accumulation from the liver and skeletal muscle to adipose tissue, which is in line with the reduced skeletal muscle mass decline observed in T2DM patients [103,106]. However, toxicology studies in monkeys showed that high-dose vildagliptin can increase the serum levels of creatine kinase (an indicator of skeletal muscle degeneration), although this response may be species-restricted [107]. The GLP-1 analog liraglutide produced a reduction in fat mass and increase in fat-free mass in obese TD2M patients after 12 weeks of treatment [108]. In contrast, a recent meta-analysis of randomized trials concluded that the GLP-1 analog semaglutide reduced both fat mass and fat-free mass [90,109]. Murine studies showed that semaglutide was effective in a male C57BL/6 mouse HFD model of sarcopenic obesity via modulation of the skeletal muscle metabolome to increase pathways related to amino acids instead of lipids and organic acids [110]. The GLP-1 analog dulaglutide has also recently shown promise as a promoter of skeletal muscle function in aged (24-month-old) C57BL/6J male mice and as an inhibitor of inflammation-associated myofiber injury in a male db/db mouse model of diabetic sarcopenia [111,112].

In summary, the T2DM drugs sulfonylureas and thiazolidinediones may not be suitable to treat sarcopenia due to documented negative effects on skeletal muscle. It may also be advisable to monitor the development of muscle atrophy in patients prescribed these drugs. The T2DM drugs metformin, SGLT2 inhibitors, DPP-IV inhibitors, and GLP-1 analogs may have the potential to treat aging-related skeletal muscle atrophy, but the reported clinical and pre-clinical data are currently inconclusive and require further investigation in models of muscle aging (Table 2).

### 2.3. Non-Coding RNAs as Potential Drug Targets

Non-coding RNAs (ncRNAs) are RNAs that are not translated into proteins and possess biological functions. The ncRNAs can be broadly divided into two classes: small RNAs (such as microRNAs, piRNAs, and scaRNAs) and long RNAs (including Xist and HOTAIR) [113]. Numerous studies have linked ncRNAs to aging-induced skeletal muscle atrophy. From a drug discovery perspective, modulation of the activity of these ncRNAs can be a strategy for developing therapeutics. This can be achieved at different points in the pathways that regulate ncRNAs. For example, the biogenesis of the ncRNA of interest could be downregulated or its function can be inhibited. ncRNA replacement could also be employed to suppress biological activity [114]. Targeting ncRNA biogenesis would be a suitable approach for developing small molecule-based drugs, whereas functional inhibition or replacement could be achieved by AntimiRs, miRNA sponges, miRNA masks, or miRNA mimics [114]. These drug development strategies are only applicable for ncRNAs that promote, rather than ameliorate, aging-related skeletal muscle atrophy. Some recent, prominent reports of ncRNA activity in sarcopenia are discussed below (a recent review is also provided by Lee and Kang [115]).

There is increasing appreciation that the regenerative capacity of satellite cells becomes compromised with aging. Shao et al. investigated the miRNA types differentially expressed in small extracellular vesicles (sEVs) released from myotubes derived from aged mice with muscle atrophy [116]. Compared to normal myotubes, the myotubes from aged mice secreted sEVs with increased levels of miR-690. This miRNA was also found to be elevated in human skeletal muscle aging [116]. miR-690 treatment was shown to inhibit satellite cell proliferation by targeting the myocyte enhancer factor 2 group of transcription factors that are known to promote myogenesis [116,117]. miR-690 silencing in aged mice alleviated muscle atrophy and enhanced satellite cell proliferation [116].

Synaptopodin-2 intron sense-overlapping lncRNA (SYISL) is known to regulate myogenesis in mice [118]. Jin et al. validated SYISL expression in humans and pigs and demonstrated that SYISL knockdown alleviated muscle atrophy in murine models of sarcopenia [119]. SYISL promoted muscle atrophy via the sponging of particular miRs via contain complementary binding sites, such as miR-23a-3p, which are known to reduce the activity of atrogenes MuRF-1 and atrogin-1 [119,120]. Overexpression of human SYISL in the skeletal muscle of aged mice accelerated mass loss through conserved atrophy pathways regulated by the master atrogene forkhead box O3 (FoxO3a) that lies upstream of MuRF-1 and atrogin-1 [119,121].

These examples of ncRNAs upregulated in conditions of muscle atrophy (such as miR-690 and SYISL) may be utilized as potential targets for candidate drugs that reduce their biogenesis or functional activity to mitigate sarcopenia progression. This strategy was recently validated by Lui et al. for the microRNA (miR)-29b, which was shown to be involved in multiple types of muscle atrophy [122]. Using virtual screening, an in-house library of 3000 compounds was assessed for binding interactions with the well-characterized three-dimensional structure of the hairpin precursor pre-miR-29b [123]. From the ten best hits, the five-membered ring compound TGP-29b-066 was most effective at specifically reducing miR-29b expression in C2C12 myotubes [123]. TGP-29b-066 prevented myotube atrophy using the angiotensin II, dexamethasone, and TNF-α treatment models, along with downregulating atrogin-1 and MuRF-1 expression [123]. This research could pave the way for future small-molecule drug development that specifically targets atrophy-related ncRNAs.

### 2.4. Development of Protein Therapeutics

Protein-based drugs have been successful in the clinic for the treatment of various diseases, including cancer, multiple sclerosis, and infections (reviewed in [124]). Examples include Bevacizumab (also known as Avastin), which is a humanized IgG for VEGF that is used to treat cancer, and Etanercept (also known as Enbrel), an Fc-conjugated TNFR2 extracellular domain protein used to treat immune diseases. However, new methodological developments may be necessary to overcome some challenges associated with protein drugs, such as resistance and individual variations to therapy [124]. Nevertheless, over 200 genuine and modified therapeutic proteins have been approved for clinical use that generate sales in excess of USD 100 billion [124].

Protein-based drugs are under development that may have applications for sarcopenia. Recently, Lee et al. built upon their previous studies of the myokine, meteorin-like (METRNL), which was shown to be released from infiltrating immune cells after skeletal muscle injury to promote myogenesis [125,126]. In aged mice (24–27 months old), adeno-associated virus (AAV)-mediated METRNL delivery via tail vein administration or recombinant rMETRNL treatment via intramuscular (IM) injection enhanced skeletal muscle regeneration after chemical injury with barium chloride [126]. Mechanistically, single-cell RNA analysis demonstrated that METRNL triggered tumor necrosis factor (TNF)-dependent apoptosis of fibro/adipogenic progenitor cells (FAPs) in the regenerating muscle, producing a microenvironment that favors myogenesis over fibrosis [126]. As a further study, it would be interesting to assess whether rMETRNL therapy can prevent the myofiber atrophy and fiber-type switching that occurs in aging muscle without chemical-induced injury, because FAPs are also implicated in the increased fibrosis, fatty infiltration, and low-grade inflammation associated with aging muscle [127].

Growth differentiation factor 11 (GDF11), also known as bone morphogenetic protein 11 (BMP-11), is a member of the TGF-β family, with an identical structure in humans and rodents [128]. GDF11 is a homolog of myostatin, and both can activate the ActRII receptor that blocks skeletal muscle growth [129]. It should be noted that there are some reports that GFD11 administration can rejuvenate skeletal muscle, although these have been considered to be controversial (discussed in [130]). Jin et al. assessed the effects of GDF11 propeptide-Fc (GDF11PRO-Fc), which inhibits GDF11 and myostatin signaling, on skeletal muscle in adult mice [129,131] Human GDF11PRO-Fc was delivered by AAV-packaging and IM injection or tail vein delivery. In adult mice, GDF11PRO-Fc IM delivery produced regions of localized myofiber hypertrophy and increased grip strength [131]. In the mdx model of muscular dystrophy, tail vein delivery produced myofiber hypertrophy, reduced fibrosis, increased grip strength, and improved endurance [131]. As a next step, it would be interesting to test the effect of GDF11PRO-Fc treatment on myofiber atrophy and functional decline in aged mice. It should also be noted that whilst GDF11PRO-Fc is a promising protein therapeutic for muscle atrophy, high levels of GDF11 in some diseases are associated with improved prognosis, such as pancreatic cancer [132].

In summary, there have been interesting recent developments in protein therapeutics for skeletal muscle atrophy. Although these have yet to be fully tested in models of aging-related muscle atrophy, the mechanisms involved (enhanced satellite cell function or ActRII receptor blockade) suggest potential for sarcopenia drug development (Table 3).

### 2.5. Natural Products and Extracts

Natural products are defined as molecules produced by living organisms and, for drug development, are usually restricted to organic compounds and secondary metabolites [133]. Natural products may possess some advantages compared to synthetic molecules because they accumulate inside living cells, such as reduced side effects [134]. Alternatively, therapeutics from natural sources may be prepared as extracts, which can be sourced in large numbers from plant materials and include both primary and secondary metabolites, such as proteins, fats and oils, dietary fibers, carbohydrates, and antioxidants [135]. Some recent advances in the development of natural compounds for treating sarcopenia are described below.

D-allulose is a naturally occurring monosaccharide sugar mainly produced from corn that possesses 70% of the sweetness of sucrose and is inert in energy metabolism [136]. D-allulose has been shown to have numerous health properties, such as anti-oxidative, anti-obesity, and neuroprotective effects [136]. Kim et al. fed 48-week-old male C57BL/6J mice with the AIN-93G diet (which promotes weight gain and hepatosteatosis similar to the HFD [137]) containing allulose for 12 weeks [138]. Allulose-fed mice showed increased quadriceps muscle mass and grip strength compared to controls [138]. RNA sequencing analysis revealed that allulose affected the expression of more than 40 genes related to protein synthesis in the gastrocnemius skeletal muscle, including downregulated myostatin and upregulated insulin-like growth factor-1 (IGF-1; a major promoter of skeletal muscle hypertrophy [139]) [138]. Additionally, allulose was shown to alleviate dysregulated autophagy and increase antioxidant enzyme activity [138].

Hesperidin is a flavanone glycoside that occurs in citrus fruits and is involved in plant defense [140]. Hesperidin is known to possess both antioxidant and anti-inflammatory properties [141]. Oh et al. assessed the effect of hesperidin on skeletal muscle aging in female C57BL/6J 22–26-month-old mice via 5 mg/kg/day oral administration for 8 months [142]. Increased mass and CSA of the quadriceps and gastrocnemius muscles were observed in the treated mice [142]. Hesperidin also increased grip strength and reduced inflammation, as shown by flow cytometry assessment of the M1 (inflammatory)/M2 (regenerative) macrophage ratio [142]. Western blotting demonstrated that hesperidin ameliorated muscle atrophy by activating the hypertrophy-associated IGF-1/Akt/mTOR signaling pathway [142].

Luteolin is a flavone compound and the principal yellow molecule from the plant *Reseda luteola* (yellow weed) that is known to produce antioxidant and anti-obesity effects in liver and adipose tissue [143,144]. Kim et al. assessed the effect of luteolin supplementation on sarcopenic obesity in the C57BL/6J murine HFD model [145]. After 20 weeks of feeding, luteolin suppressed lipid infiltration into the gastronemius muscle and downregulated inflammatory factors, atrogenes, and p38 mitogen-activated protein kinase (MAPK) activity [145]. These effects produced a reduction in protein degradation and improvement in muscle function measured via grip strength.

Fisetin is a flavonol compound found in fruits and vegetables. Interestingly, fisetin has been identified as a senolytic agent that selectively kills senescent cells [146]. In light of the increased numbers of senescent cells found in aging tissues, Liu et al. tested fisetin in the Z24−/− mouse model of Progeria syndrome (premature aging) via oral gavage for 4 weeks [147]. Fisetin treatment reduced the number of senescent FAPs, which influence the coordination of skeletal muscle regeneration [148], and increased the number of muscle stem cells in the gastrocnemius muscle [147]. Muscle pathology was also improved, as demonstrated by increased myofiber CSA and reduced fibrosis [147].

The plant-based Mediterranean diet is rich in olive oil and has been consistently linked with reduced rates of noncommunicable diseases, such as heart disease and cancer [149]. Although this diet has been shown to produce beneficial effects on skeletal muscle mass, there is no evidence of a positive effect on sarcopenia [150]. González-Hedström et al. further investigated components of the olive plant as possible treatments for sarcopenia by focusing on the leaf extract that is known to be rich in phenolic compounds [151]. Thus, 24-month-old Wistar rats (equivalent to 60 years in humans) were treated with 100 mg/kg olive leaf extract dissolved in drinking water for 21 days. Extract treatment increased protein content in the gastrocnemius muscle and downregulated myostatin expression. Insulin sensitivity was increased in the gastrocnemius via activation of the PI3K-Akt pathway and anti-inflammatory effects [151].

Coffee silver skin (the thin layer that covers coffee seeds) is known to possess antioxidant and neuroprotective properties [152,153]. With the aim of characterizing novel natural products with the potential to treat sarcopenia, Kim et al. applied activity-guided fractionation to coffee silver skin extracts [154]. Only the ethanol extract was effective at myostatin inhibition in a cell-based assay using HEK293 cells. Further, 6-week-old ICR outbred male mice were orally administered the ethanol extract diluted in saline for 29 days, which produced increases in grip strength and forelimb muscle mass (calculated by subtracting bone mass from the weight of the forelimbs). Solvent partitioning, chromatography, and HPLC were used in conjunction with the myostatin inhibition assay to characterize two active compounds in the extract: *^β^N*-arachinoyl-5-hydroxytryptamide and *^β^N*-behenoyl-5-hydroxytryptamide [154].

As an alternative to the plant-derived natural products described above, Okamura et al. tested Brazilian green propolis (a resinous mixture produced by honey bees to build hives) in a model of sarcopenic obesity [155]. Propolis has previously been shown to improve lipid metabolism, insulin resistance, and obesity in humans [156]. The db/db T2DM and obesity mouse model was fed the chow diet supplemented with propolis powder from 8 to 16 weeks of age. Propolis-fed mice showed greater grip strength and increased mass of the soleus (slow twitch type I) and plantaris (fast twitch type II) muscles [155]. The accumulation of saturated fatty acids was also inhibited in the soleus muscle, along with reduced expression of genes associated with inflammation and atrophy [155]. The beneficial effect of propolis on sarcopenic obesity was linked with improvements in the gut microbiota, which is known to influence skeletal muscle physiology (reviewed by Liu et al. [157])

Farnesol is a natural acyclic sesquiterpene alcohol and building block for steroid precursors in plants, animals, and fungi [158]. Bae et al. identified farnesol as a potent inducer of PGC-1α expression by high-throughput screening in C2C12 myoblasts [159]. Oral administration of farnesol to female 26-month-old C57BL/6J mice for 1 month produced increases in soleus muscle force, stronger grip strength, and enhanced energy expenditure [159]. Parkin-interacting substrate (PARIS/*Zfp746*) is a transcriptional repressor of PGC-1α that was found to be upregulated in aged skeletal muscle. Farnesol treatment induced PARIS farnesylation (a type of prenylation) that alleviated PARIS-mediated PGC-1α suppression in the aging muscle [159].

Computational approaches have been applied to natural-product-based drug discovery for sarcopenia. In a recent example, Ali et al. used computational screening to identify small-molecule inhibitors of myostatin in a library of approximately 38,000 compounds from a traditional Chinese medicine database [160]. Two compounds (termed ZINC85592908 and ZINC85511481) were identified based on high binding affinity and specificity. The authors recommended ZINC85592908 for further development as a myostatin signaling inhibitor due to its greater stability [160]. Similarly, Ahmad et al. screened 2000 natural compounds for myostatin inhibition using the freely available PatchDock algorithm [161]. Dithymoquinone (a bioactive compound from the flowering plant *Nigella sativa*) showed the most favorable binding free energy. Molecular dynamics analysis indicated that dithymoquinone blocks the myostatin-dependent activation of the ActRIIB receptor [161].

Overall, these recent examples of natural product development, from plant or animal sources, show that this continues to be a productive research area for discovering novel therapeutics for sarcopenia (Table 4). Computational-based approaches can also be utilized to identify natural-product-based candidates for further validation in cell-based and animal models.

### 2.6. Drug Repositioning

A significant advantage of drug repositioning compared with traditional drug discovery methodologies is that the repositioned compound has already been characterized in other disease context(s) and clinical trials [162]. A number of successes using this approach have confirmed its public health benefits and market value [163]. Additionally, high-profile novel sarcopenia drug development failures in the pharmaceutical industry have generated interest in drug repositioning as an alternative approach [164]. The anti-diabetes drugs tested in models of skeletal muscle atrophy described above can also be considered as examples of repositioning. Drugs developed for disorders other than diabetes have also been tested as repositioning candidates. Some recent examples are described below.

Losartan is a hypertension medication that works by blocking activation of the angiotensin II receptor. Heart failure patients treated with losartan also show improved skeletal muscle function [165]. Kang et al. tested the effect of losartan in a rat model of sarcopenia. Male 4-week-old F344xBN rats were administered 0.6 g/L losartan diluted in drinking water over a period of 18 months [166]. Average muscle mass of the tibialis anterior (TA), soleus, and peroneus longus muscles was increased by losartan treatment combined with an exercise program, compared to exercise alone [166]. Interestingly, soleus (slow twitch type I) and peroneus longus (fast twitch type II) muscle mass was also increased without the need for an exercise program. Protein concentration assessment showed elevated levels of mTOR, Akt, and extracellular signal-regulated kinase (ERK, a branch of MAPK signaling [167]) in all muscles analyzed [166].

Lithium (Li) is a gold-standard therapy for bipolar disorder and has been shown to prevent muscle cell death in a cell-based model of oculopharyngeal muscular dystrophy [168]. However, Li is also associated with a number of side effects and toxicity burden [169]. One of the biological targets of Li is inositol monophosphatase (IMPase). Lee at al. investigated the effects of ebselen, an IMPase inhibitor with reduced toxicity, in a glycerol treatment model that models the skeletal muscle fat cell accumulation observed in sarcopenia [170,171]. Ebselen improved muscle performance, as measured by latency to fall in the rotarod test, increased myofiber CSA in the gastrocnemius muscle, and downregulated atrogin-1 and MuRF-1 expression [172].

Malotilate is a clinically safe drug developed to promote liver regeneration [173]. The biological target is arachidonate 5-lipoxygenase (Alox5), which has been implicated in numerous diseases [174]. Kim et al. treated 21-month-old male C57BL/6J mice with malotilate via oral administration for 4 weeks [175]. Malotilate produced increases in the gastrocnemius and soleus mass, along with a reduction in gonadal adipose tissue mass. Grip strength was improved, and immunohistochemical analysis showed an increase in the CSA of type II myofibers. RNA sequencing revealed an upregulation of IGF-1 in murine C2C12 myotubes treated with malotilate [175].

Trametinib (brand name Mekinist) is an orally bioavailable drug used to treat melanoma by the inhibition of dual specificity mitogen-activated protein kinase kinase 1 (MEK1) and MEK2 [176]. The Raf–MEK–ERK phosphorylation cascade is also involved in myostatin and GDF11 signaling. Masuzawa et al. treated aged mice with trametinib and observed downregulated ERK activity in skeletal muscle [177]. This was accompanied by increased myofiber CSA compared to the vehicle-treated aged mice [177].

Etanercept (brand name Enbrel) is a biopharmaceutical drug used to treat rheumatoid arthritis, plaque psoriasis, and ankylosing spondylitis [178]. It is a fusion protein of the TNF receptor to the constant end of the IgG1 antibody. Sciorati et al. treated 16-month-old female C57/BL6 mice with weekly doses of etanercept delivered subcutaneously until the mice reached 28 months old [179]. Etanercept improved skeletal muscle performance, as measured by the hanging wire test, and increased myofiber CSA in the gastrocnemius muscle. Etanercept significantly increased the proportion of type II myofibers, and immunohistochemical analysis of CD3 (a lymphocyte marker) and CD68 (a macrophage marker) indicated that etanercept treatment also reduced muscle inflammation [179].

Computational-based methodologies have been applied to drug repositioning for sarcopenia. A recent example was provided by Liang et al. based on the analysis of skeletal muscle transcriptomic sequencing data in humans and mice to obtain the Gene Signature of Sarcopenia, coupled with literature validation [180]. This led to the identification of vorinostat (brand name zolinza, a histone deacetylase inhibitor used to treat cutaneous T cell lymphoma [181]) as the top-ranking drug. The potential of vorinostat to treat sarcopenia was assessed in murine C2C12 myotubes. Vorinostat increased myotube differentiation, diameter, and MHC levels [180].

The above studies of drug repositioning for sarcopenia show that different drug classes with varying mechanisms of action have the potential to be effective for this disorder (Table 5). An effective method for identifying repositioning candidates is to select drugs that target mechanistic pathways which overlap with the original disease application and the development of sarcopenia.

## 3. New Drug Targets

Due to the lack of approved drugs for sarcopenia, a pivotal aspect of drug development is the characterization of new targets and related pathways. There have been a number of recent papers describing novel factors that may be applicable for drug development. These are summarized in Table 6.

## 4. Future Challenges

Sarcopenia drug development is challenging, and there is no clinically approved drug for this disease. However, the potential global market value for sarcopenia therapeutics was estimated at USD 2.71 billion in 2021 and is expected to reach USD 3.9 billion in 2028 due to factors such as demographic aging in developed countries [191]. The major challenges associated with sarcopenia drug development include the complexity of aging-related conditions in older persons and issues that may confound the parameters of physical frailty and sarcopenia, such as sarcopenic obesity [192].

From the pre-clinical standpoint presented in this review, there is no shortage of new drug candidates and target pathways reported in the recent research literature. One issue is the identification of those drug candidates that would be most likely to succeed in clinical trials. This may especially be an issue for academic research, where there may not always be the incentive for researchers to carry their work to the ‘next step’ of commercialization. One solution may be to further build collaborations between academia and industry, or the utilization of ‘technology institutes’ within academic institutions to provide support and guidance on issues such as patenting and technology transfer to companies [193].

Another potential problem impeding drug development is our incomplete understanding of the disease process. As mentioned above, sarcopenia is a complex, multifactorial disorder. Even in animal models, these disease mechanisms are not yet fully characterized. For example, a number of studies covered in this review used atrogin-1 and MuRF-1 downregulation as an indicator of drug effectiveness. However, a recent integrated genomic and proteomic analysis of different types of murine skeletal muscle atrophy models by Hunt et al. showed that protein levels of atrogin-1 and MuRF-1 were not increased, and mRNA levels were only upregulated in glucocorticoid or cancer-induced muscle atrophy but not in atrophy caused by aging [194]. The authors demonstrated that protein levels of the lysosomal protease cathepsin L were upregulated in all three types of muscle atrophy and may serve as a more robust marker of drug effectiveness. Ongoing basic research is still required to increase our understanding of the disease processes underlying aging-related skeletal muscle atrophy and to provide reliable markers for drug development.

This review focused on recent reports of sarcopenia drug candidates showing effectiveness in cell-based and animal models. It should be noted that compounds which have been reported to extend lifespan may also prevent sarcopenia by countering the aging process in multiple tissues. One prominent example is the mTOR inhibitor rapamycin that prevents aging-related skeletal muscle loss in mouse models [195]. Recently, MyMD-1 (a synthetic derivative of alkaloid myosmine that inhibits TNF-α production) was shown to extend lifespan in male and female 19-month-old C57BL/6 mice [196]. As part of the analysis, MyMD-1 increased muscle strength in the treated mice. Thus, there is an overlap between sarcopenia research and more generalized areas of aging, such as frailty. It may be noted that cheap and simple indices of frailty have been developed for murine studies that show high correlation with assessments of human aging [197,198]. These could be utilized in sarcopenia drug discovery to ascertain whether the therapeutic of interest is primarily affecting skeletal muscle or diverse tissues affected by the aging process.

The majority of studies discussed in this review relied on previous reports of bioactivity (for example, anti-inflammation or antioxidant effects) followed by assessment in cell and animal models of muscle aging. There appear to be relatively few reports of cell-based high-throughput screening protocols for hit compound identification from chemical libraries. One recent attractive example is the C2C12 myoblast PGC-1α expression system originally reported by Kim et al. and followed-up by Bae et al. [159,199]. This allowed for the rapid identification of two U.S. Food and Drug Administration-approved library compounds with potential anti-sarcopenia activity (indoprofen and farnesol). It can be envisaged that developing myoblast-based screening systems with expression constructs for other key regulators of muscle atrophy, such as AMPK, p70S6 kinase, or FoxO3a, would also be useful for the rapid selection of primary hit compounds.

Bioinformatics-based approaches are being increasingly utilized to identify new candidate drugs and targets. However, as mentioned above, these drugs are not always validated by detailed assessment in animal models. This could provide an opportunity for increased collaboration between ‘dry’ bioinformatics and ’wet’ skeletal muscle labs to fully validate whether these new discoveries are suitable for clinical development.

Our survey of the literature revealed that the majority of new drugs undergoing pre-clinical investigation are based on previously reported compounds. There was a relative lack of novel chemical entities or structures being developed for anti-sarcopenia activity. One recent example of novel compound development is a 16-mer d-peptide by Takayama et al. that showed myostatin inhibitory activity [200]. An arginine-containing derivative of the peptide (termed MID-35) was directly injected into the TA muscle of 8-week-old male C57BL/6J mice. Thus, 28 days later, the TA muscle mass was significantly increased compared to the vehicle-injected TA. As a next step, it would be interesting to assess the effect of single-dose MID-35 delivery into the TA of aged mice.

Currently, the aged mouse (at least 18 months old or, ideally, older [201]) can be considered as the ‘gold standard’ of animal model validation for anti-sarcopenia drug candidates. However, there is now greater appreciation of the complexity of aging-related skeletal muscle atrophy in humans. For example, sarcopenia progression can be influenced by other aging-related disorders, such as diabetes, heart failure, chronic obstructive pulmonary disease (COPD), Alzheimer’s disease, obesity, or CKD [202]. It may be rewarding to additionally test candidate drugs in these models to gain further insight into their potential to treat muscle wasting in the aged population. Example models that also show muscle atrophy include the db/db type 2 diabetes mouse, intranasal elastase model of COPD, and the HFD model of obesity.

Although the aged mouse is a preferred experimental model of sarcopenia, there can be logistical problems associated with its use. From the authors’ own experiences, aged mice are expensive to order from suppliers and are often in limited supply. Establishing a colony of aged mice from young adults is also time consuming and expensive due to animal housing costs. As an initial test of drug potential to treat sarcopenia, it may be preferable to utilize alternative and readily available genetic models of aging, such as the senescence-accelerated mouse P8 (SAMP8) or the Zmpste24−/− accelerated aging model of progeria syndrome, as described by Liu et al. and Guo et al. [147,203].

As a final consideration, there is increasing appreciation that the highly complex and multifactorial pathogenesis of sarcopenia will require therapeutic approaches to be multimodal [204]. Therefore, rather than relying on a single drug treatment to alleviate sarcopenia, it would be beneficial to assess novel candidates in the context of nutritional and/or exercise protocols that have been previously established in aging models.

## 5. Summary and Concluding Remarks

Sarcopenia is a challenging disease for drug development, and there is currently no clinically approved therapeutic. Outcomes in clinical trials depend on functional gains in muscle performance, rather than just increases in mass, while also being well tolerated with low side effects. Sarcopenia is also a complex multifactorial disorder, and the underlying mechanisms are not fully understood. This review focused on pre-clinical drug development for sarcopenia. Due to the lack of approved therapeutics and a large projected market value, there are a large number and variety of different compounds and target pathways/cellular mechanisms under investigation. A large proportion of current research is focusing on natural compounds and extracts, due to their characterized biological activity and advantages for further drug development. Much research effort is also focusing on the role of ncRNAs in sarcopenia progression, which can provide targets for small molecules currently under development for inhibiting ncRNA biogenesis. A number of type 2 diabetes drugs, such as SGLT2 inhibitors, DPP-IV inhibitors, and GLP-1 analogs, are also being investigated for their effects on skeletal muscle mass in T2DM patients and animal models. It will be important to consider whether these drugs can also be effective in the context of pre-diabetes or normoglycemia. Mitochondria have a pivotal role in maintaining muscle function and are known to become dysfunctional in aging. Mitochondria-targeting drugs also hold great promise for treating sarcopenia and may utilize recent advances in mitochondria drug delivery systems [205]. Drug repositioning strategies are also providing clinically validated candidates with known pharmacokinetics in humans. These previously characterized drugs can also provide new insights into the molecular pathways regulating skeletal muscle atrophy. A wider adoption of cell-based screening systems, based on known master regulatory genes, such as PGC-1α, could accelerate throughput and increase the number of hits for further analysis. Overall, much effort is being focused on identifying drug candidates with promising pre-clinical therapeutic activity in sarcopenia models, which raises the probability of successful drug development for this debilitating and increasingly prevalent disease.

## Figures and Tables

**Figure 1 cells-12-02608-f001:**
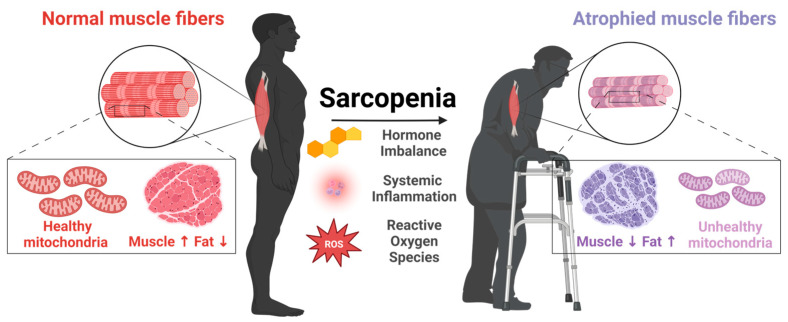
Aging-related changes in skeletal muscle structure and function.

**Figure 2 cells-12-02608-f002:**
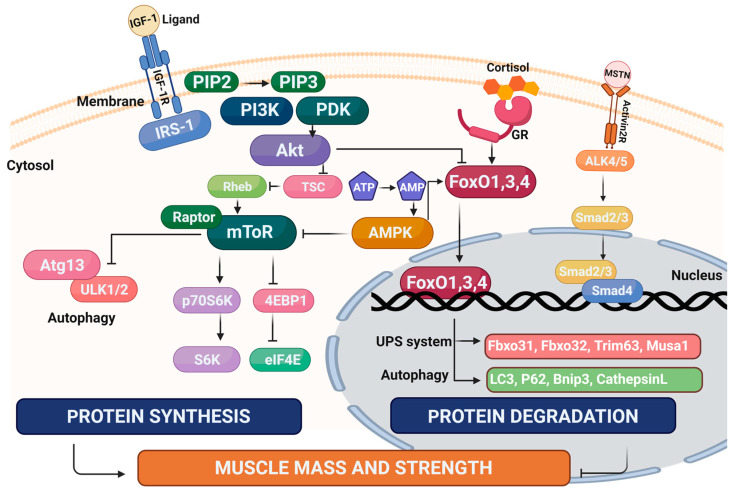
The major sarcopenia-related signaling pathways in myofibers. Arrows designate pathway activation and bars designate inhibition. Target genes that are linked to skeletal muscle wasting are also shown. Abbreviations: MSTN: Myostatin, IGF-1: Insulin growth factor-1, IRS-1: Insulin receptor substrate 1, PI3K: Phosphoinositide 3-kinase, mToR: mammalian target of rapamycin, P70S6K: Ribosomal protein S6 kinase beta-1, 4EBP1: Eukaryotic translational initiation factor 4E-binding protein 1, AMPK: AMP-activated protein kinase, ALK4/5: Activin receptor type-1B, FoxO1,3,4: Forkhead box O1,3,4, PDK: 3-phosphoinositide-dependent protein kinase-1, TSC: Tuberous sclerosis 1, Rheb: Ras homolog enriched in brain, ATG13: Autophagy-related protein 13, Ulk1/2: Unc-51-like autophagy-activating kinases 1 and 2, eIF4E: Eukaryotic translational initiation factor 4E.

**Table 1 cells-12-02608-t001:** Mitochondria-targeting compounds.

Drug	Target/Mechanism	Experimental System	Further Development?	Ref.
Elamipretide (SS-31)	Reduce ROS and stabilize cardiolipin	Female C57BL/6Nia mice (26 mo)	Endurance tested (no grip strength)	[68]
BAM15	Lowers respiratory coupling and anti-obesity	Male C57BL/6 mice (80 wo) with HFD (10 weeks)	Suitable for sarcopenic obesity	[70]
MRL-45696	Pan PARP inhibitor	Male C57BL/6 mice (10 wo) with HFD (5 days)	Endurance tested (no grip strength)	[72]
Mitoquinone mesylate (MitoQ)	Synthetic antioxidant analogue of coenzyme Q10	Male CD2F1 C26 tumor bearing mice (11 wo)	Not yet tested in aged mice	[74]
Mito-TEMPOL	Superoxide dismutase mimetic	(1) Male ICR(CD-1) mice with CLP surgery (2) Male C57BL/6J mice with CKD surgery	Not yet tested in aged mice	[76,77,78]

**Table 2 cells-12-02608-t002:** Anti-diabetes drugs.

Drug	Target/Mechanism	Experimental System	Further Development?	Ref.
Metformin	Involvement of the AMPK pathway	T2DM patients and mice	Inconclusive results	[91,92,93,94,95,96]
Sulfonylurea, meglitinide/ glinides	ATP-sensitive K^+^ channel inhibitors	Meta analysis in patients	Not suitable	[100]
Pioglitazone, lobeglitazone	PPAR activators	T2DM patients	Not suitable	[101,102,103]
Glibenclamide	SGLT2 inhibitor	T2DM patients	Inconclusive results	[105,106]
Vildagliptin	DPP-IV inhibitor	T2DM patients and monkeys	Inconclusive results	[107,108]
Liraglutide, semaglutide, dulaglutide	GLP-1 analogs	T2DM patients, HFD-fed or aged mice	Inconclusive results	[109,110,111,112,113]

**Table 3 cells-12-02608-t003:** Protein therapeutics.

Drug	Target/Mechanism	Experimental System	Further Development?	Ref.
METRNL	Enhances myogenesis	(1) BaCl_2_ TA muscle injury model (2) Aged male C57BL/6J mice (24–26 mo)	Not assessed in aged mice without injury	[127]
GDF11PRO-Fc	GDF11 and myostatin inhibitor	(1) Adult male mice the (2) mdx model of Duchenne muscular dystrophy	Not assessed in aged mice	[132]

**Table 4 cells-12-02608-t004:** Natural products and extracts.

Drug	Target/Mechanism	Experimental System	Further Development?	Ref.
D-allulose	Anti-oxidative, anti-obesity, and neuroprotective effects	Male C57BL/6J mice (48 wo) with AIN-93G diet (12 weeks)	Not assessed in aged mice	[139]
Hesperidin	Antioxidant and anti-inflammation	Female C57BL/6J mice (22–26 mo)	Suitable	[143]
Luteolin	Antioxidant and anti-obesity	Male C57BL/6J mice (5 wo) with HFD (20 weeks)	Not assessed in aged mice	[146]
Fisetin	Senolytic agent	Male Zmpste24^−/−^ (Z24^−/−^) mice (4 mo)	Not assessed in naturally aged mice	[148]
Olive leaf extract	Reduced rates of noncommunicable diseases	Elderly aged over 65 yo; Male Wistar rats (24 mo)	Functional data required	[151,152]
Coffee skin silver skin extracts	Antioxidant and neuroprotection	Male ICR mice (6 wo)	Not assessed in aged mice	[155]
Brazilian green propolis	Improves lipid metabolism, reduces insulin resistance	Male db/db mice (8 wo)	Not assessed in aged mice	[156]
Farnesol	Inducer of PGC-1α	Female C57BL6/J mice (23–26 month)	Suitable	[160]

**Table 5 cells-12-02608-t005:** Drug repositioning.

Drug	Target/Mechanism	Experimental System	Further Development?	Ref.
Losartan	Angiotensin II receptor inhibitor	Male F344xBN rats (4 wo)	Functional test required	[167]
Ebselen	IMPase inhibitor	Male C57BL/6J mice (13 wo): Dexamethasone and glycerol atrophy models	Not assessed in aged mice	[171]
Malotilate	Alox5 inhibitor	Male C57BL/6J mice (21 mo)	Suitable	[176]
Trametinib	MEK1 and MEK2 inhibitor	Male C57BL/6J mice (24 mo)	Functional test required	[178]
Etanercept	TNF signalling inhibitor	Female C57BL/6J mice (28 mo)	Endurance tested (no grip strength)	[180]
Vorinostat	Histone deacetylase inhibitor	C2C12 myotubes	Not assessed in aged mice	[181]

**Table 6 cells-12-02608-t006:** Recently reported factors linked to sarcopenia that may be applicable for drug development.

Factor	Role	Experimental System	Reference
Platelet glycoprotein 4 (CD36)	Fatty acid import and metabolism; Inflammatory responses	Geriatric assessment of frail older humans	[182]
*N*- Oleoylethanolamide	Circulating and peripheral endocannabinoid system	Aged rats	[183]
Eukaryotic translation initiation factor 4E-binding protein 1 (4EBP1)	mTOR complex 1 (mTORC1) regulation of mRNA translation	Transgenic mouse strains (TSC1mKO, S6K1mKO, S6K1-TSC1mKO, 4EBP1mt-muscle and 4EBP1mt-TSC1mKO)	[184]
Interferon-inducible guanylate-binding protein	Cell-autonomous immunity	Senescence-accelerated mouse model	[185]
Ubiquitin protein ligase E3 component n-recognin 4	E3 ubiquitin-protein ligase	*Drosophila melanogaster* and aged mice	[186]
Hydroxyprostaglandin dehydrogenase 15-(NAD) (15-PGDH)	Regulation of prostaglandin expression	Aged mice	[187]
TNF superfamily member 12	Facilitation of apoptosis	Mendelian Randomization in aged Europeans	[188]
Hepatocyte growth factor	Paracrine regulation of cell growth, motility and morphogenesis	Mendelian Randomization in aged Europeans	[188]
α-Klotho	Regulation of TGF-β signaling	Aged mice	[189]
Leukocyte immunoglobulin-like receptor subfamily B member 2 (CD85)	Antigen-dependent immune response	Mendelian Randomization of sarcopenia-related traits (UK Biobank)	[190]
Asporin	Cartilage matrix	Mendelian Randomization of sarcopenia-related traits (UK Biobank)	[190]
Contactin-2	Cell adhesion	Mendelian Randomization of sarcopenia-related traits (UK Biobank)	[190]
Ecto-ADP-ribosyltransferase 4	Unknown (ADP-ribosylation not detected)	Mendelian Randomization of sarcopenia-related traits (UK Biobank)	[190]
Superoxide dismutase 2	Clearance of mitochondrial ROS	Mendelian Randomization of sarcopenia-related traits (UK Biobank)	[190]

## Data Availability

Data sharing not applicable. No new data were created or analyzed in this study.

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
