# Peer review of "Age Is Just a Number: Progress and Obstacles in the Discovery of New Candidate Drugs for Sarcopenia"

_cells, 2023, doi:10.3390/cells12222608_

Round 1
Reviewer 1 Report
Comments and Suggestions for Authors
Please, see attached file.

Author Response
Response to Reviewer comments for the manuscript ‘Age is just a number: Recent progress in the discovery of new candidate drugs for sarcopenia.’ Manuscript ID 2704788 (Cells)’
Reviewer 1 Comments to Author:
Dear Authors,
Please, consider and assess the following comments and suggestions.
There are a few minor typescripts that should be amended:
Our response: We apologize for typescripts in our manuscript. We have amended the text accordingly and indicated the changes using red text.
In addition, there is lack of references in the text. Following there are a few of them, but the manuscript must be reviewed thoroughly, and every statement must be properly referenced.
Our response: Thank you very much for these comments about the manuscript references. We have inserted the references at the suggested points in the text. In addition, we have revised the manuscript to ensure that the statements are properly referenced. The inserted references are shown using red text.
We wish to convey our gratitude to Reviewer 1, whose insightful comments have enhanced the quality of our manuscript.

Reviewer 2 Report
Comments and Suggestions for Authors
Te review is very comprehensive, well written and has merit. I only have some suggestions:
The main question addressed is effect of exercise on sarcopenia in cancer.
The topic is relevant, however, the review is very general, not enough specific and it sounds like a textbook text.This review in current form does not add anything new to the published material. Therefore it needs substantial improvements, especially being more specific about different types of exercise linked to specific molecular pathways. It also should point out the specific gaps in current knowledge about every discussed aspect of the review.
The authors should consider performing systematic review instead of just narrative review.
In current form of the manuscript there are no specific conclusions.
The references are too numerous, and especially for the introductory parts of manuscript too general and not specific to the main question addressed.
The figures are too general. There should be more specific figures outlining molecular mechanisms etc. Also a table summarising the existing evidence in systematic fashion should be made as is usual for systematic review.
I suggest to add a comprehensive summary table of all the drugs reviewed, where main target of action is epmphasized and the level of evidence (cell model, animal model, human clinical trials) is stated and pointed out what is not yet known about effects of this drug in sarcopenia.
Author Response
Response to Reviewer comments for the manuscript ‘Age is just a number: Recent progress in the discovery of new candidate drugs for sarcopenia.’ Manuscript ID 2704788 (Cells)’
Reviewer 2 Comments to Author:
Te review is very comprehensive, well written and has merit. I only have some suggestions:
The main question addressed is effect of exercise on sarcopenia in cancer.
The topic is relevant, however, the review is very general, not enough specific and it sounds like a textbook text. This review in current form does not add anything new to the published material. Therefore it needs substantial improvements, especially being more specific about different types of exercise linked to specific molecular pathways. It also should point out the specific gaps in current knowledge about every discussed aspect of the review.
The authors should consider performing systematic review instead of just narrative review.
In current form of the manuscript there are no specific conclusions.
Our response: Thank you very much for this advice about our manuscript. We have revised the abstract to state more clearly the different types of candidate drugs that will be discussed. In addition, to provide a summary and conclusions for the drugs discussed in the main review text, we have added tables for each type of drug that summarize the existing research progress and what remains to be assessed for further development as a treatment for sarcopenia. The new tables are indicated using red text.
The references are too numerous, and especially for the introductory parts of manuscript too general and not specific to the main question addressed.
Our response: We apologize for the excessive number of references in the introductory parts of the manuscript. For the manuscript revision, we have deleted unnecessary references where possible and checked that the remaining references are related to the topic being addressed.
The figures are too general. There should be more specific figures outlining molecular mechanisms etc. Also a table summarising the existing evidence in systematic fashion should be made as is usual for systematic review.
I suggest to add a comprehensive summary table of all the drugs reviewed, where main target of action is epmphasized and the level of evidence (cell model, animal model, human clinical trials) is stated and pointed out what is not yet known about effects of this drug in sarcopenia.
Our response: Thank you for these comments about the figures and tables. In the revised manuscript, the figure outlining molecular mechanisms now includes the target genes that are known to be upregulated in skeletal muscle wasting. As mentioned above, for the revision we have included tables for each type of drug discussed in the review, which states the evidence obtained and what remains to be investigated as a candidate for sarcopenia. These revisions are indicated using red font.
We wish to convey our gratitude to Reviewer 2, whose insightful comments have enhanced the quality of our manuscript.

Reviewer 3 Report
Comments and Suggestions for Authors
The review is devoted to the problem of sarcopenia and possible drug candidates to slow it down. The review is written in an interesting way, focusing on unexplored issues and possible directions for solving them. The review draws attention to the fact that the nature and mechanisms of development of sarcopenia are unknown. There are currently no recommended drugs for its treatment. More research has been done on animals and tissue culture and very little work has been done on human subjects. In this regard, in my opinion, the title of the review is too promising for something that does not yet exist. It makes sense to change it somewhat in accordance with the real state of affairs in the development of treatments for sarcopenia.
In the abstract, I recommend indicating what types of drugs will be examined in the review (for example, by the title of the sections of the review).
The experiments with mice only are described In the chapter «Mitochondria targeting compounds». It makes sense to consider here ways to increase the oxidative potential in humans. Such possibilities and drugs are described in sports physiology.
The review may be useful to a wide range of researchers and students.
Author Response
Response to Reviewer comments for the manuscript ‘Age is just a number: Recent progress in the discovery of new candidate drugs for sarcopenia.’ Manuscript ID 2704788 (Cells)’
Reviewer 3 Comments to Author:
The review is devoted to the problem of sarcopenia and possible drug candidates to slow it down. The review is written in an interesting way, focusing on unexplored issues and possible directions for solving them. The review draws attention to the fact that the nature and mechanisms of development of sarcopenia are unknown. There are currently no recommended drugs for its treatment. More research has been done on animals and tissue culture and very little work has been done on human subjects. In this regard, in my opinion, the title of the review is too promising for something that does not yet exist. It makes sense to change it somewhat in accordance with the real state of affairs in the development of treatments for sarcopenia.
Our response: We are very grateful for this advice about the manuscript title. To make the title sound less promising, we have changed it to ‘Age is just a number: progress and obstacles in the discovery of new candidate drugs for sarcopenia.’ This change is indicated using red text.
In the abstract, I recommend indicating what types of drugs will be examined in the review (for example, by the title of the sections of the review).
Our response: Thank you for this comment. To indicate what types of drugs will be examined in the review, the following text has been added to the abstract:
‘The types of drugs examined include mitochondria targeting compounds, anti-diabetes agents, small molecules that target non-coding RNAs, protein therapeutics, natural products and repositioning candidates.’
(shown using red text).
The experiments with mice only are described In the chapter «Mitochondria targeting compounds». It makes sense to consider here ways to increase the oxidative potential in humans. Such possibilities and drugs are described in sports physiology.
Our response: Thank you very much for this suggestion. Accordingly, the following text has been added to the ‘Mitochondria targeting compounds’ chapter:
‘The studies described above indicate that strategies to reduce oxidative stress and maintain the mitochondrial network in skeletal muscle may be beneficial to prevent aging-related weakness. In this regard, indicators from human studies in research fields such as sports physiology could be useful for selecting compounds that reduce oxidative stress in aging muscle (see Braakhuis and Hopkins for a review[1]). For example, the antioxidant N-acetylcysteine has been shown to delay skeletal muscle fatigue in healthy subjects and intravenous administration in athletes produced improvements in sprint performance [2, 3]. In addition, spirulina (an extract of blue-green algae containing numerous antioxidants) is believed to have been consumed by the Cuban and Chinese Olympic teams to enhance the response to training [1, 2].’
(shown using red text).
The review may be useful to a wide range of researchers and students.
We wish to convey our gratitude to Reviewer 3, whose insightful comments have enhanced the quality of our manuscript.
References for response to Reviewers’ comments:
- Braakhuis, A.J. and W.G. Hopkins, Impact of Dietary Antioxidants on Sport Performance: A Review. Sports Med, 2015. 45(7): p. 939-55.
- Matuszczak, Y., et al., Effects of N-acetylcysteine on glutathione oxidation and fatigue during handgrip exercise. Muscle Nerve, 2005. 32(5): p. 633-8.
- Cobley, J.N., et al., N-Acetylcysteine's attenuation of fatigue after repeated bouts of intermittent exercise: practical implications for tournament situations. Int J Sport Nutr Exerc Metab, 2011. 21(6): p. 451-61.
